# Calibration of Collision Recovery Coefficient of Corn Seeds Based on High-Speed Photography and Sound Waveform Analysis

**Xinping Li \*, Wantong Zhang, Shendi Xu, Fuli Ma, Zhe Du, Yidong Ma and Jing Liu**

College of Agricultural Equipment Engineering, Henan University of Science and Technology, Luoyang 471000, China; 210321041644@stu.haust.edu.cn (W.Z.); ztwtant@163.com (F.M.); lk35eee@163.com (Z.D.); qllzp4@163.com (Y.M.); 210321041647@stu.haust.edu.cn (J.L.)
\* Correspondence: lwzyyxs@163.com; Tel.: +86-13592065522

**Abstract:** Compared with the movement of corn seeds in the sowing machine, the movement in the threshing machine is more intense. The collision between corn seeds and threshing parts and other corn seeds will not only change the movement path of corn seeds in the threshing clearance but also cause damage to corn seeds. Therefore, when using discrete element simulation technology to optimize the critical components of corn threshing machinery, it is necessary to measure corn seeds' accurate collision recovery coefficient. However, when measuring the collision recovery coefficient between corn seeds, there will be multi-point collisions between corn seeds, affecting the measurement results' accuracy. In order to solve this problem, this study combined high-speed photography and the sound waveform of corn seed collision to eliminate the interference of the multi-point collision phenomenon and improve the accuracy of measurement results. According to the above test method, the contact parameters of corn seeds were measured. Finally, the corn–corn rolling friction coefficient and corn–PMMA rolling friction coefficient were 0.0784 and 0.0934, respectively. The corn–corn static friction coefficient was 0.32, and the corn–PMMA static friction coefficient was 0.445. The corn–corn collision recovery and corn–PMMA collision recovery coefficients were 0.28 and 0.62, respectively. After that, the method's reliability and the measurement results' accuracy were verified by the plane collision test and repose angle test.

**Keywords:** corn seeds; collision recovery coefficient; static friction coefficient; mult-point collision; sound waveform; characteristic dimensions

## 1. Introduction

Corn is grown in many parts of the world and has the highest planting area and yield in China. According to China's National Bureau of Statistics, the country's corn output in 2021 was 272.55 million tons, and the sown area was 433.24 million hectares. The sowing and harvesting of corn are the two most critical links in the mechanization of corn production [1–4]. However, due to the complex external environment, it is impossible to clearly observe the movement and stress of corn in the above machinery, which can not provide favorable information for the optimization of sowing or harvesting machinery. Discrete element simulation technology can simulate the stress and movement of corn in the above machinery and provide a theoretical reference for the structural design and optimization of critical components [5–9].

The most commonly used software is EDEM when using discrete element technology to optimize corn sowing and harvesting machinery. When using EDEM for simulation, it is necessary to set corn seeds' intrinsic and contact parameters [10–13]. The intrinsic parameters of corn are consistent with those obtained by physical experiments. However, due to the difference between the shape of the generated corn seed model and the actual corn seed, there are errors between the contact parameters of corn seeds and the

actual measured values, so it is necessary to calibrate the contact parameters of corn seeds. According to the energy conservation theory and high-speed photography technology, Cui et al. [14] calculated the rolling friction coefficient between corn seeds and polymethyl methacrylate, steel, and corn seeds. Han et al. [15] glued rice onto a plate that could adjust the angle and made a particle plate. Then, the coefficient of static friction between rice was measured using the inclined plane method. Wang et al. [16] calibrated the rolling and static friction coefficients between corn seeds by combining physical and simulation tests. Based on the repose angle test data, a mathematical regression model obtained the rolling and static friction coefficients between corn seeds. Li et al. [17] used the Hertz–Mindlin with JKR Cohesion contact model to calibrate the relevant parameters of the clayey black soil with two moisture contents through the repose angle simulation test. By combining physical tests and simulation tests, Yu et al. [18] obtained the collision recovery, static, and rolling friction coefficients between panax notoginseng seeds. Through physical tests, Xing et al. [19] obtained several contact parameters of latosol particles in Hainan hot areas. Then, they applied the Plackett–Burman and Box–Behnken tests to obtain the best combination of parameters. When the response surface optimization method is used to calibrate the particle contact parameters, there is a problem in that the calibration parameters are distorted due to improper selection of factor zero level. To solve this problem, Zhang et al. [20] established a linear function of the zero level of variable range and the measured value during factor calibration. They obtained the best combination of static and rolling friction coefficients between corn seeds through the repose angle simulation test. When calibrating the contact parameters of camellia seeds, Ding et al. [21] adopted a BP artificial neural network based on a genetic algorithm to optimize the contact parameters.

Currently, most research on the discrete element parameter calibration of corn seeds is to optimize the critical components of corn sowing machinery through discrete element simulation technology. In sowing machinery, the movement of corn seeds is mainly rolling and sliding. The lifting method is the most commonly used in measuring the repose angle of corn seeds. In the process of using the lifting method, the motion of corn seeds is also mainly rolling and sliding, and the collision between corn seeds is relatively rare. Therefore, the rolling and static friction coefficients of corn seeds significantly influence the repose angle. When the optimum contact parameters of corn seeds obtained using the above method are applied to the discrete element simulation of the sowing machinery, the simulation results can maintain a high consistency with the physical tests. However, in the working process of the threshing machine, the movement of corn seeds is more intense, and the corn seeds will collide with the threshing parts and other corn seeds when it falls off the corn cob. These collisions will not only affect the movement of corn seeds in the threshing clearance but also cause damage to corn seeds. Therefore, in the study of low-damage threshing machinery, it is a very effective means to use discrete element simulation technology to analyze the stress and movement of corn seeds in threshing machinery to optimize its critical components. So, due to the above phenomena, the collision recovery coefficient of corn seeds will significantly impact the simulation results in the simulation process. Therefore, we need to measure corn seeds' exact collision recovery coefficient to improve the simulation results' accuracy.

The methods to measure the collision recovery coefficient of corn seeds include the inclined plane (plane) impact method and the single pendulum impact method. When the collision recovery coefficient between corn seeds is measured using the inclined plane (plane) collision method, the multi-point collision occurs due to the irregular surfaces of corn seeds, which reduces the accuracy of the measurement results. When the single pendulum collision method is used to measure the collision recovery coefficient between corn seeds, it is necessary to fix two corn seeds on the rope, then release one corn seed from a certain height to make it collide with another corn seed, and finally calculate the collision recovery coefficient between corn seeds according to the position changes of the two corn seeds. However, during the experiment, the tension of the rope on the corn seed will affect

the measurement results, and the collision point of the two corn seeds will be artificially interfered with, which reduces the randomness of the experiment to a certain extent.

In order to solve the above problems, this study randomly selected a certain number of corn seeds. Then, these corn seeds were divided into three categories according to their surfaces and defined and measured the characteristic dimensions of these three categories of corn seeds. After measuring the characteristic dimensions of these three categories of corn seeds, the classification and statistical results of corn seeds are modified according to the relationship between the characteristic dimensions of corn seeds in order to establish an accurate discrete element simulation model. Then, the inclined plane method measured corn seeds' static friction coefficient. The energy conservation method based on high-speed photography measured corn seeds' rolling friction coefficient. The inclined plane collision method measured corn seeds' collision recovery coefficient. When measuring the collision recovery coefficient of corn seeds, the multi-point collision of corn seeds is removed by combining the high-speed photography technology and the sound waveform of the collision to improve the accuracy of the measurement results. Finally, the reliability of the measurement method and the accuracy of the measurement results are verified by the simulation and physical test of the plane collision test and repose angle test of corn seeds.

## 2. Materials and Methods

### 2.1. Materials

This study used the corn variety Zhengdan 958 to measure the contact parameters. The measurement of corn seed contact parameters includes the contact parameters between corn seeds and the contact parameters between the corn seeds and working parts. The material of the working part is polymethyl methacrylate (PMMA), so the contact parameters to be measured are the corn–corn static friction coefficient, the corn–corn rolling friction coefficient, the corn–corn collision recovery coefficient, the corn–PMMA static friction coefficient, the corn–PMMA rolling friction coefficient, and the corn–PMMA collision recovery coefficient. When measuring the contact parameters between corn seeds, it is necessary to use a particle plate made of corn seeds (as shown in Figure 1). When testing the contact parameters between corn seeds, the particle plate is installed on the test device to determine the contact parameters.

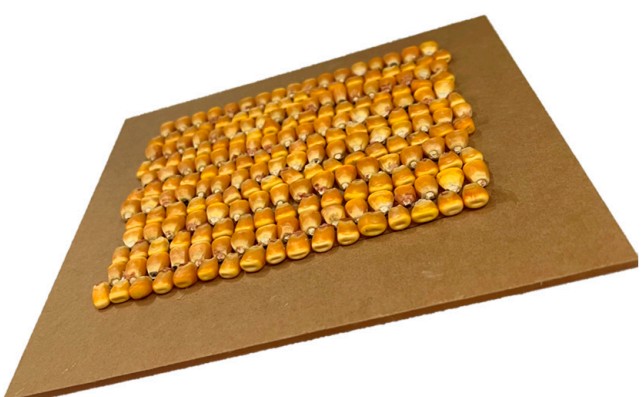

**Figure 1.** Particle plate.

### 2.2. Methods

In this study, the proposed measurement method's reliability and the measurement results' accuracy were verified by the plane collision test and repose angle test.

In the plane collision test, the particle plate made in advance was placed on the table, and a corn seed was selected to fall freely from a certain height. The high-speed camera was used to shoot the falling process of the corn seed and record the maximum height of the corn seed after the first collision (Figure 2a). The lifting method was used to measure the

repose angle of corn seeds, and the test device is shown in Figure 2b. Place the bottomless PMMA cylinder on the horizontally fixed PMMA bottom plate, select six hundred corn seeds into the cylinder, and lift the PMMA cylinder at a speed of 0.04 m·s$^{-1}$.

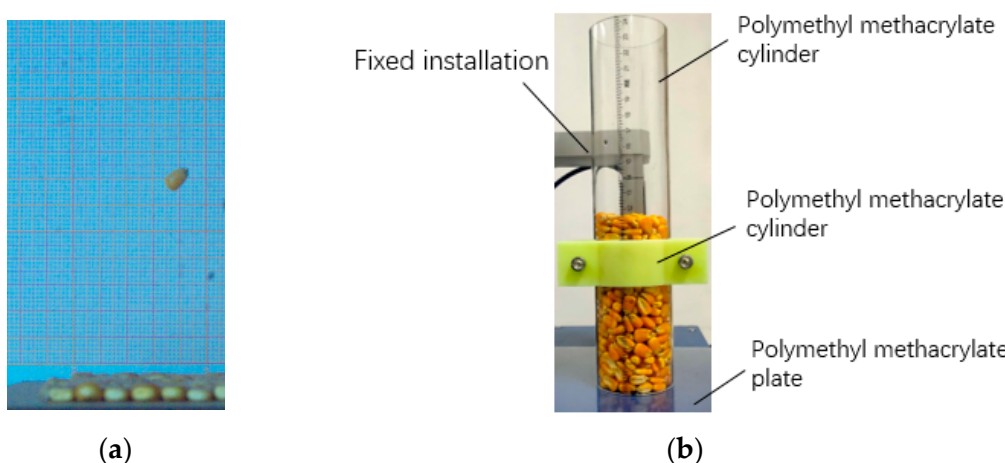

(**a**)                      (**b**)

**Figure 2.** Validation test: (**a**) Plane collision test; (**b**) Test device for repose angle test.

### 3. Results and Discussion

*3.1. Determination of Intrinsic Parameters of Corn Seeds*

3.1.1. Density and Moisture Content of Corn Seeds

The corn seed variety was Zhengdan 958, which was selected for the experiment, and the density of the corn seeds was 1197 kg·m$^{-3}$, measured via the drainage method. The moisture content is measured according to the material moisture measurement method specified in the national standard GB/T3543.6. One hundred grams of corn seeds are randomly selected and dried in a constant temperature drying oven. The weight of the corn seeds after drying is measured, and the moisture content of the corn seeds is calculated by formula (1):

$$\omega = \frac{m_1 - m_2}{m_1} \times 100\% \tag{1}$$

where $\omega$ is the moisture content; $m_1$ is the mass of the corn seeds before drying and $m_2$ is the mass of the corn seeds after drying.

The above process was repeated five times, and the average moisture content of the three groups of corn seeds was 11.7%.

3.1.2. Determination of Poisson's Ratio and Shear Modulus

Poisson's ratio of corn seed was 0.4, referring to the literature [22]. The shear modulus of corn seeds is measured using a universal testing machine and calculated according to the following formula:

$$\begin{cases} E = \frac{FL_1}{S(L_2 - L_1)} \\ K = \frac{E}{2(1 + \mu)} \end{cases} \tag{2}$$

where $E$, $K$, and $\mu$ are the elastic modulus of corn seed, shear modulus of corn seed, and Poisson's ratio of corn seed; $F$ and $S$ are the maximum bearing capacity of corn seed at elastic deformation stage and the cross-sectional area at the center of the corn seed; $L_1$ and $L_2$ are the lengths of the corn seeds before and after compression.

A uniaxial compression test was carried out on corn seeds using a universal testing machine, and the impact location of corn seeds is shown in Figure 3. Repeat the experiment three times and calculate the average. The shear modulus of corn seed is $1.36 \times 10^8$ Pa.

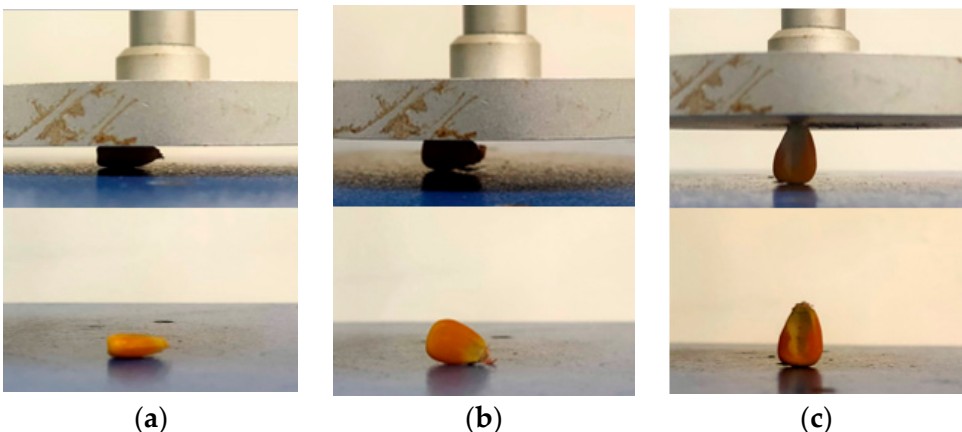

**Figure 3.** Impact locations of corn seeds: (**a**) impact the ventral part of the corn seed; (**b**) impact the side of the corn seed; (**c**) impact the top surface of the corn seed.

### 3.1.3. Determination of Characteristic Dimensions of Corn Seeds

Because the surfaces of corn seeds significantly influence the contact parameters of corn seeds [23], the selected corn seeds are divided into flat, quasi-conical, and quasi-cylindrical shapes according to their surfaces, and the characteristic dimensions of these three types of corn seeds are measured. The definition method of corn seed characteristic dimensions is shown in Figure 4.

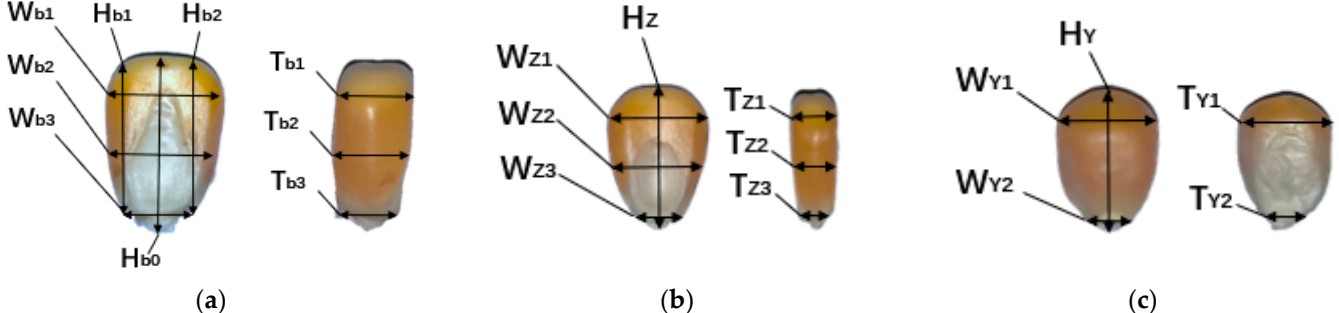

**Figure 4.** Characteristic dimensions definition diagram of corn seed: (**a**) flat; (**b**) quasi-conical; (**c**) quasi-cylindrical.

Three thousand corn seeds were randomly selected and classified according to the above shapes. The proportion of flat, quasi-conical, and quasi-cylindrical corn seeds was 75.7%, 17.6%, and 6.7%, respectively. The corn seeds were measured according to the outline dimensions of the corn seeds defined in Figure 4. Two hundred flat and two hundred quasi-conical corn seeds were randomly selected for measurement. All quasi-cylindrical corn seeds were measured, and the measurement results are shown in Table 1.

As can be seen from the measurement results in Table 1, the ratio between $W_{b3}$ and $W_{b2}$ of flat corn seeds is 0.553, and the ratio between $W_{Z3}$ and $W_{Z2}$ of quasi-conical corn seeds is 0.485. This study takes 0.5 as the standard value. The corn seed is flat when the ratio between the width of the bottom end of the corn seed and the width of the middle position is greater than or equal to 0.5. Otherwise, it is quasi-conical. According to the above conclusions, the corn seeds were classified again, and the proportion of each type of corn seed was counted. The proportion of flat, quasi-conical, and quasi-cylindrical corn seeds was 77.9%, 15.4%, and 6.7%, respectively. The modification results of characteristic dimensions of corn seeds are shown in Table 2.

**Table 1.** Summary of corn seed characteristic dimensions measurement results.

| Shape | Characteristic Dimensions | Mean Value (mm) |
|---|---|---|
| Flat | $W_{b1}$ | 8.21 |
| | $W_{b2}$ | 7.77 |
| | $W_{b3}$ | 4.30 |
| | $H_{b0}$ | 12.77 |
| | $H_{b1}$ | 11.56 |
| | $H_{b2}$ | 11.50 |
| | $T_{b1}$ | 4.17 |
| | $T_{b2}$ | 4.29 |
| | $T_{b3}$ | 3.76 |
| Quasi-conical | $W_{Z1}$ | 7.80 |
| | $W_{Z2}$ | 7.21 |
| | $W_{Z3}$ | 3.50 |
| | $H_Z$ | 5.07 |
| | $T_{Z1}$ | 4.83 |
| | $T_{Z2}$ | 3.10 |
| | $T_{Z3}$ | 12.23 |
| Quasi-cylindrical | $W_{Y1}$ | 7.05 |
| | $W_{Y2}$ | 6.54 |
| | $H_Y$ | 4.08 |
| | $T_{Y1}$ | 3.62 |
| | $T_{Y2}$ | 10.28 |

**Table 2.** Modification results of characteristic dimensions of corn seeds.

| Shape | Characteristic Dimensions | Mean Value (mm) |
|---|---|---|
| Flat | $W_{b1}$ | 8.14 |
| | $W_{b2}$ | 7.83 |
| | $W_{b3}$ | 4.37 |
| | $H_{b0}$ | 12.71 |
| | $H_{b1}$ | 11.57 |
| | $H_{b2}$ | 11.45 |
| | $T_{b1}$ | 4.14 |
| | $T_{b2}$ | 4.29 |
| | $T_{b3}$ | 3.78 |
| Quasi-conical | $W_{Z1}$ | 7.76 |
| | $W_{Z2}$ | 7.20 |
| | $W_{Z3}$ | 3.32 |
| | $H_Z$ | 5.02 |
| | $T_{Z1}$ | 4.80 |
| | $T_{Z2}$ | 3.04 |
| | $T_{Z3}$ | 12.18 |
| Quasi-cylindrical | $W_{Y1}$ | 7.05 |
| | $W_{Y2}$ | 6.54 |
| | $H_Y$ | 4.08 |
| | $T_{Y1}$ | 3.62 |
| | $T_{Y2}$ | 10.28 |

*3.2. Determination of Static and Rolling Friction Coefficient of Corn Seeds*

3.2.1. Static Friction Coefficient

The static friction coefficient is measured using the inclined plane sliding method [24,25], and the measurement principle is shown in Figure 5. Place the corn seed on a test plate in a horizontal position, slowly increase the angle between the test plate and the horizontal

plane and record the angle when the corn seed appears to slide. When the corn seed first appears to slide, the force balance equation on the test plate is as follows:

$$\begin{cases} F_1 = G\sin\alpha = f \\ F_2 = G\cos\alpha = N \\ \quad f = \mu F_2 \end{cases} \tag{3}$$

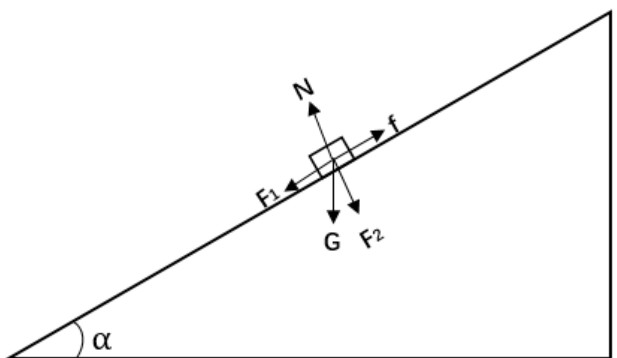

**Figure 5.** Principle diagram of static friction coefficient test.

The formula for calculating the static friction coefficient of corn seeds on the test plate is

$$\mu_1 = \frac{f}{F_2} = \frac{F_1}{F_2} = \frac{mg\sin\alpha}{mg\cos\alpha} = \tan\alpha \tag{4}$$

where $F_1$, $F_2$, $f$, and $N$ are tensile force, the pressure of the corn seed on the test plate, static friction force between the corn seed and the test plate and support force of test plate on corn seed; $\mu_1$ is coefficient of static friction between corn seed and test plate; $\alpha$ is angle between test plate and horizontal plane.

Figure 6 shows the contact parameter measuring device, which can be installed with different sizes and materials of the test plate. In this study, the test plate is a PMMA plate with a side length of 200 mm and a thickness of 3 mm. Place the corn seed in the center of the test plate and place the angle meter on the right side of the test plate. Slowly turn the test plate, use a high-speed camera to record the image of the moment when the corn seed appears to slide, and record the angle shown by the angle meter.

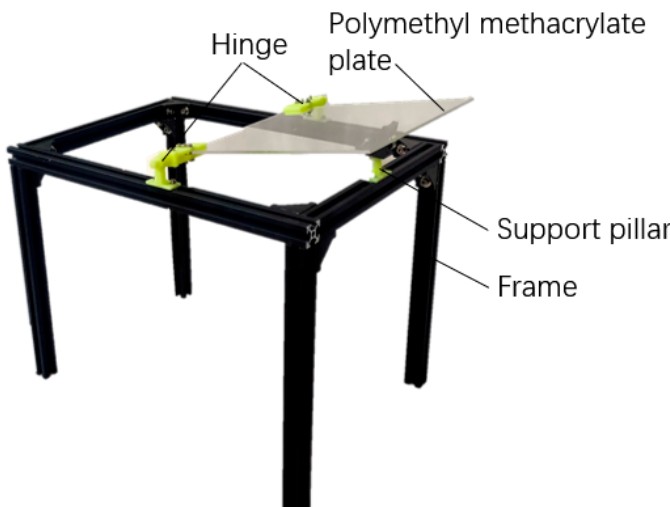

**Figure 6.** Contact parameter measuring device.

The static friction coefficient of corn–PMMA was calculated according to formula (4). When the coefficient of static friction between corn seeds is measured, the test plate is replaced with a prepared particle plate, and the test process is repeated. The two tests were repeated fifteen times, respectively, and the average value was obtained: the corn–PMMA static friction coefficient was 0.445, and the corn–corn static friction coefficient was 0.32.

### 3.2.2. Rolling Friction Coefficient

According to the energy conservation method, the rolling friction coefficient was measured using the contact parameter measuring device in Figure 6, and the measurement principle was shown in Figure 7.

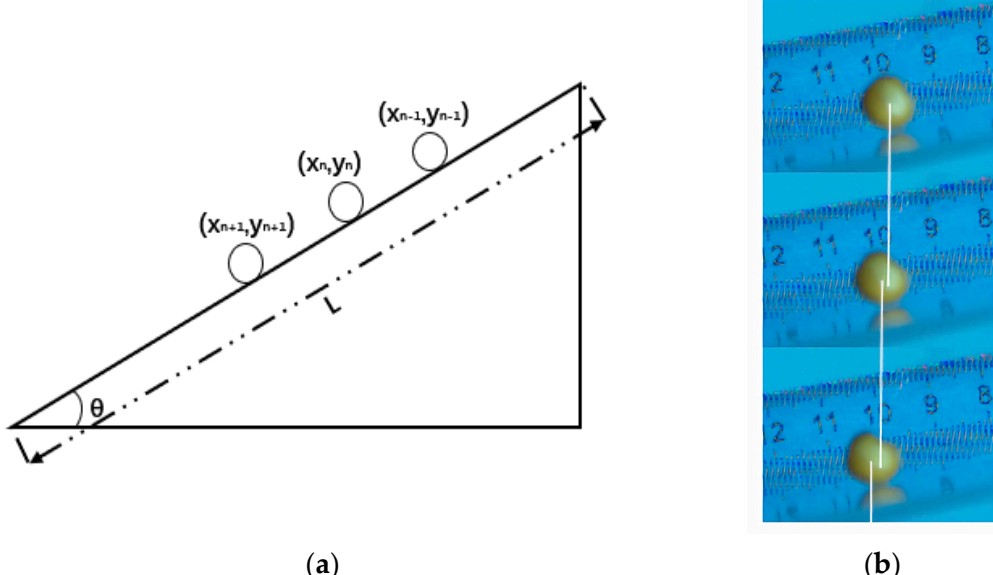

(**a**)            (**b**)

**Figure 7.** Rolling friction coefficient measurement method: (**a**) Principle diagram of rolling friction coefficient test; (**b**) The motion tracking of quasi-cylindrical corn seeds: Under the influence of the outer surface, flat and quasi-conical corn seeds display infrequent rolling. In contrast, quasi-cylindrical corn seeds are more prone to rolling. Therefore, experiments were conducted using quasi-cylindrical corn seeds. The X coordinate and Y coordinate of the corn seed are the cosine value and the sine value of the scale value of the position of the corn seed.

Fix the ruler on the test plate, and then turn the test plate to a certain angle to fix it. After that, the corn seed was placed on the test plate to make them naturally fall, and the rolling process of the corn seed was recorded with a high-speed camera (The shooting frame rate of the high-speed camera was set to 1000). The speed of the corn seed at the end of rolling is calculated according to the rolling image of the corn seed captured. Assuming that the rolling of corn seed ends at "$n$" frames, the time interval between "$n$" frames and the previous frame is 0.001 s, and the movement of corn seeds during this time interval is regarded as uniform rolling. The following formula can calculate the corn seed's speed at the rolling end:

$$V_n = \frac{k\left(\sqrt{(x_n - x_{n-1})^2 + (y_n - y_{n-1})^2} + \sqrt{(x_{n+1} - x_n)^2 + (y_{n+1} - y_n)^2}\right)}{2\Delta t} \tag{5}$$

where $V_n$, $k$, and $\Delta t$ are the speed at which the corn seed rolls at the end, the ratio factor of the actual size to the image size, and the time interval between two frames of the images.

According to the law of conservation of energy, the formula for calculating the proportion of energy lost by corn seed in the rolling process is as follows:

$$C_{f1} = \frac{U - E_k}{U} = k_1 \cot \theta_1 \tag{6}$$

where $C_{f1}$, $U$, and $E_k$ are the proportion of energy loss caused by rolling friction to total energy, gravitational potential energy at the initial position of corn seed, and kinetic energy at the end of the rolling process of corn seed. It can be seen from Formula (6) that $C_{f1}$ and $\cot \theta_1$ have a linear relationship.

The values of $C_{f1}$ and $\cot \theta_1$ are calculated by selecting ten angles from $20°\sim42°$ according to Formulas (5) and (6). Each angle is repeated fifteen times, and the average value is taken. The calculated values of $C_{f1}$ and $\cot \theta_1$ are shown in Table 3.

**Table 3.** The proportion of the energy loss caused by rolling corn seeds on the PMMA plate.

| Angle (°) | $\cot\theta_1$ | $C_{f1}$ |
|---|---|---|
| 20 | 2.747 | 0.273 |
| 22 | 2.475 | 0.251 |
| 25 | 2.145 | 0.257 |
| 27 | 1.963 | 0.181 |
| 30 | 1.732 | 0.241 |
| 32 | 1.600 | 0.165 |
| 35 | 1.428 | 0.151 |
| 37 | 1.327 | 0.131 |
| 40 | 1.192 | 0.135 |
| 42 | 1.111 | 0.137 |

According to the data in Table 3, the scatter plot of the proportion of energy loss during the rolling of corn seeds was drawn (Figure 8).

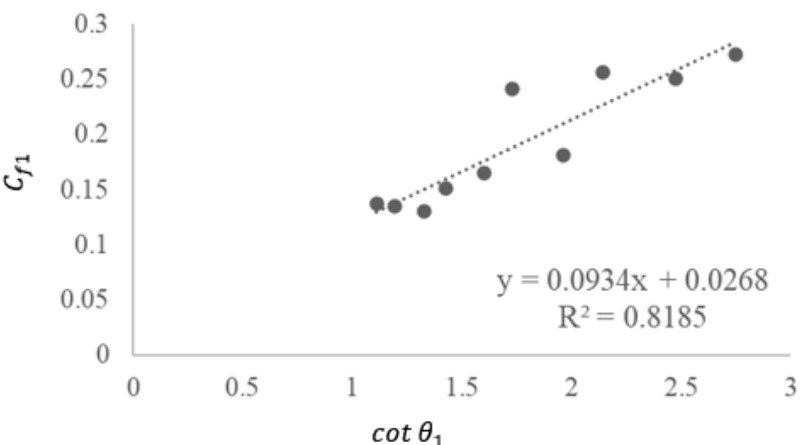

**Figure 8.** Corn–PMMA rolling friction coefficient.

The energy loss ratio $C_{f1}$ of corn seeds rolling on the PMMA plate is fitted with the cotangent value of the angle between the test plate and the horizontal plane, and the equation is obtained as follows:

$$y = 0.0934x + 0.0268 \left( R^2 = 0.8185 \right) \tag{7}$$

The rolling friction coefficient between corn seeds was measured by replacing the test plate with a prepared particle plate. The same procedure was followed to obtain the energy loss ratio table (Table 4) and the energy loss scatter diagram (Figure 9) for rolling corn seeds on the particle plate.

**Table 4.** The proportion of the energy loss caused by rolling corn seeds on the particle plate.

| Angle (°) | $\cot\theta_2$ | $C_{f2}$ |
| --- | --- | --- |
| 20 | 2.747 | 0.681 |
| 22 | 2.475 | 0.647 |
| 25 | 2.145 | 0.622 |
| 27 | 1.963 | 0.645 |
| 30 | 1.732 | 0.633 |
| 32 | 1.600 | 0.629 |
| 35 | 1.428 | 0.612 |
| 37 | 1.327 | 0.593 |
| 40 | 1.192 | 0.553 |
| 42 | 1.111 | 0.5 |

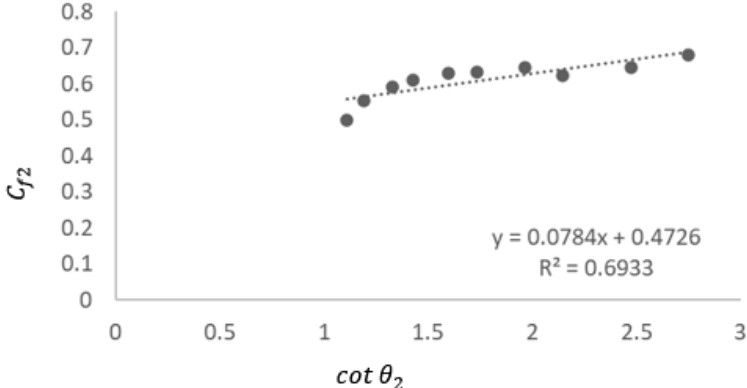

**Figure 9.** Corn–corn rolling friction coefficient.

After fitting the data, the equation is obtained as follows:

$$y = 0.0784x + 0.4726\left(R^2 = 0.6933\right) \tag{8}$$

*3.3. Determination of Collision Recovery Coefficient of Corn Seeds*

3.3.1. Multi-Point Collision of Corn Seeds

When measuring the corn–corn collision recovery coefficient, multi-point collision will occur between corn seeds due to the influence of the irregular surface of corn seeds and the angle of the collision of corn seeds. Figure 10 shows the process of single-point collision of corn seeds. Corn seeds bounce immediately after the collision (Figure 10a).

Figure 11 shows the process of the multi-point collision of corn seeds. Affected by the collision angle, the corn seed will swing or rotate after the first collision (Figure 11a), and the second collision will occur (Figure 11b). By comparing Figures 10e and 11e, it can be seen that multi-point collision between corn seeds will consume part of the kinetic energy of corn seed, thus reducing the height of corn seed's rebound and changing the motion trajectory of corn seed, reducing the accuracy of measurement results.

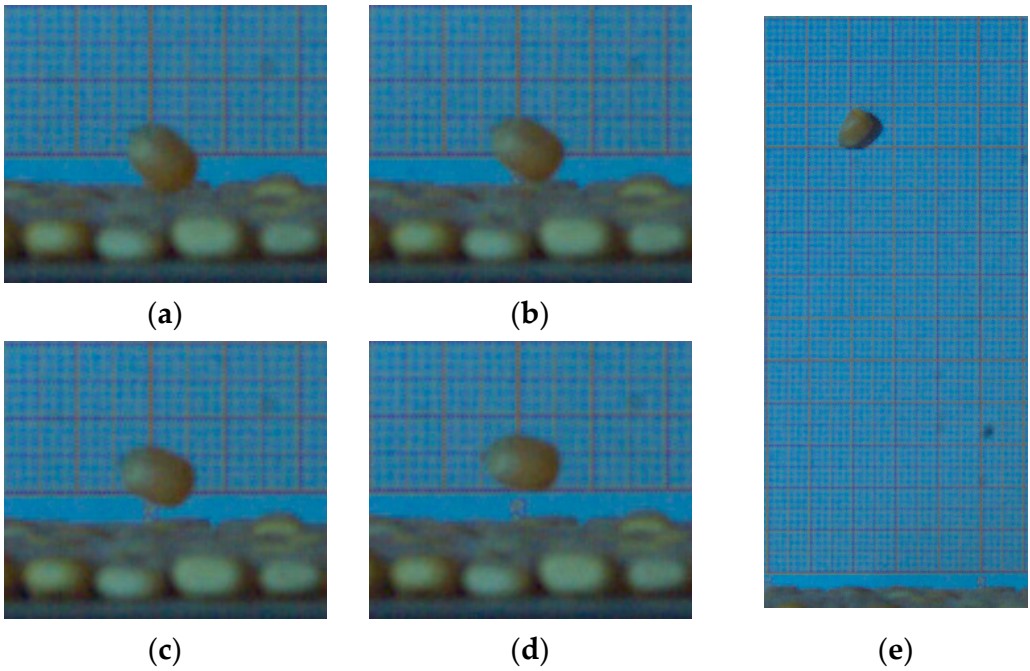

**Figure 10.** Single-point collision of corn seeds: (**a**) 1.232 s; (**b**) 1.233 s; (**c**) 1.234 s; (**d**) 1.235 s; (**e**) 1.380 s.

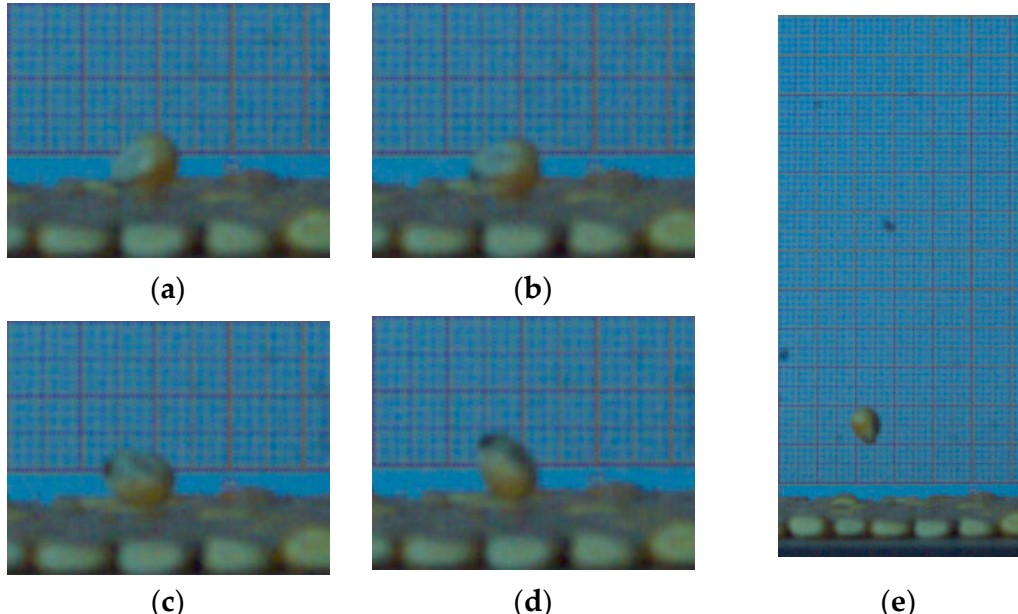

**Figure 11.** Multi-point collision of corn seeds: (**a**) 1.379 s; (**b**) 1.380 s; (**c**) 1.381 s; (**d**) 1.382 s; (**e**) 1.432 s.

In most cases, the multi-point collision can be identified in the pictures of collisions between corn seeds captured by high-speed cameras. However, due to the limitations of the structure of the test device, shooting angle, light, and the intensity of multi-point collision, some multi-point collision situations cannot be identified via high-speed photography. In order to solve this problem, this study combined high-speed photography and sound waveform to eliminate multi-point collision tests and improve the accuracy of measurement results. In the experiment, high-speed cameras were used to capture the images of corn seeds when they collided, and the sound of the collision of corn seeds was recorded. The recorded sound was then imported into Audacity, the sampling rate was selected according

to the sampling device, and the waveform was enlarged. Then, the time node of the multi-point collision was found by the recorded sound waveform, and the multi-point collision was identified according to the change of sound waveform and the captured corn seed collision pictures. Figure 12 shows the high-speed photographic images of the single-point collision of corn seeds and the waveform of the collision sound. Figure 12a is the collision between corn seeds, corresponding to the sound waveform (The collision sound of corn seeds was processed using Audacity) in the time interval of $\Delta t_1$ in Figure 12c; Figure 12b shows the rebound of corn seeds after the collision, corresponding to the sound waveform in the time interval of $\Delta t_2$ in Figure 12c.

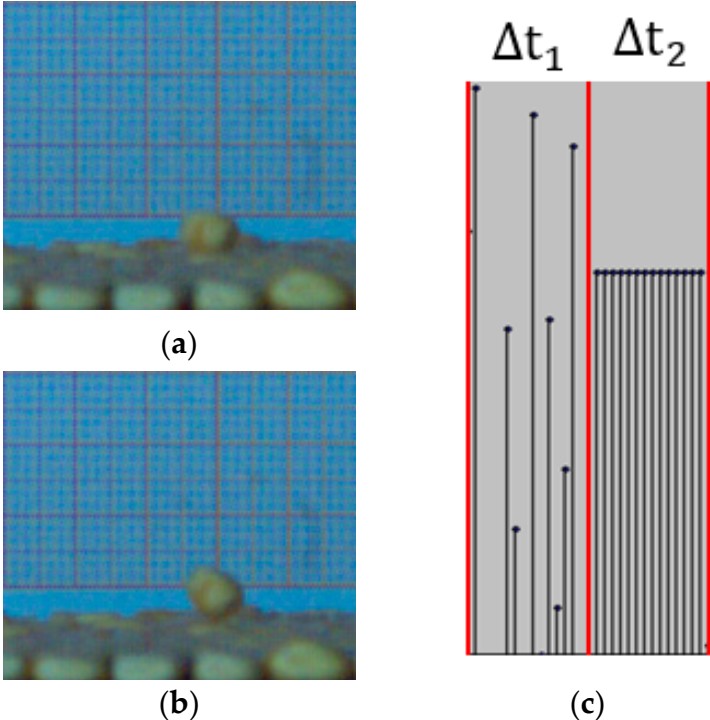

**Figure 12.** The sound waveform of the single-point collision: (**a**) Single-point collision; (**b**) the rebound of corn seeds after the collision; (**c**) sound waveforms generated by the collision between corn seeds.

Figure 13 shows the high-speed photographic images of the multi-point collision of corn seeds and the waveform of the collision sound. Figure 13a,b are two collisions of corn seeds, corresponding to the sound waveforms in Figure 13d's two-time intervals of $\Delta t_1$ and $\Delta t_2$ respectively. Figure 13c shows the rebound of corn seed after the second collision, corresponding to the sound waveform in the $\Delta t_3$ in Figure 13d.

In combination with Figures 12c and 13d, it can be seen that no matter the single-point or multi-point collision of corn seeds, each collision will cause changes in sound waveform. In the case of multi-point collision, the peak values of sound waveforms generated by different collisions were less different. After the single-point collision of corn seeds, the peak value of the sound waveform was positive and maintained the same value for a specific time interval. After the multi-point collision of corn seeds, the peak value of the sound waveform was zero and maintained the same value for a specific time interval.

The combination of high-speed photography and sound waveforms can not only identify multi-point collisions that are difficult to observe. Moreover, the time node of the first collision of corn seeds can be found faster through the sound waveform, thus reducing the time cost and improving the test efficiency.

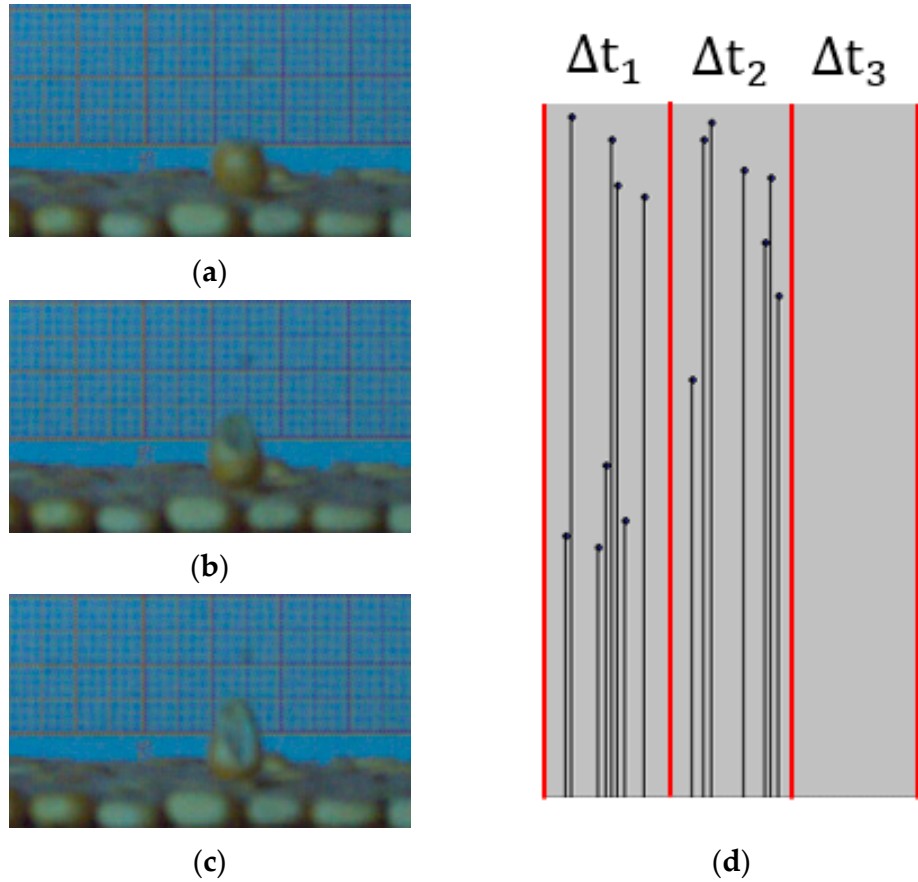

**Figure 13.** The sound waveform of the multi-point collision: (**a**) First collision; (**b**) Second collision; (**c**) The rebound of corn seeds after the second collision; (**d**) Sound waveforms generated by collisions between corn seeds.

### 3.3.2. Test Principle of Collision Recovery Coefficient

The collision recovery coefficient is measured using the inclined plane collision method [26], and the measurement principle is shown in Figure 14.

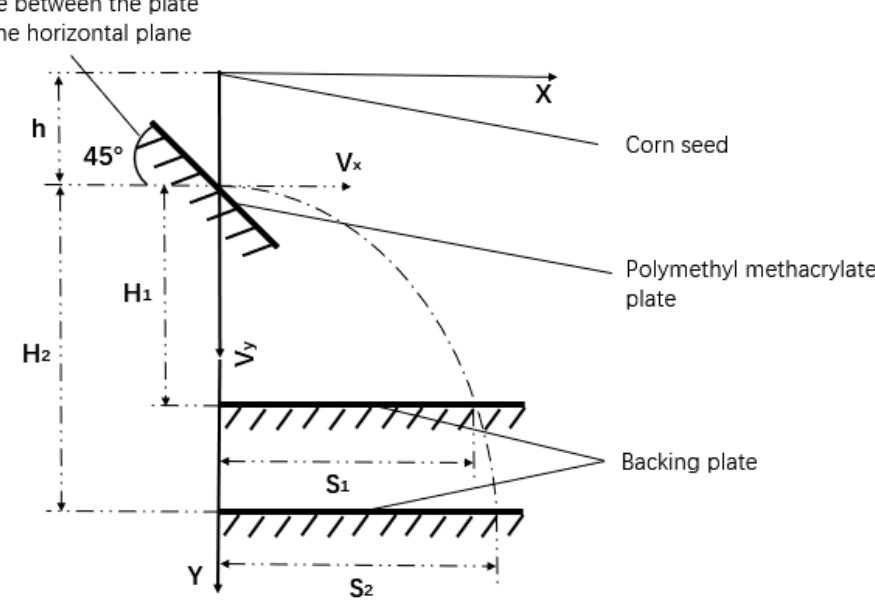

**Figure 14.** Principle diagram of inclined plate collision test.

The test plate is placed at an angle, and the corn seed falls from a certain height, collides with the test plate, and finally falls to the backing plate. The oblique motion of corn seed can be decomposed into uniform motion along the X direction and uniform motion with acceleration $g$ along the Y direction. The motion equation is as follows:

$$\begin{cases} S_i = V_x t \\ H_i = V_y t + \frac{1}{2} g t^2 \end{cases} \tag{9}$$

where $S_i$, $H_i$ ($i$= 1, 2) are the displacement in the X direction and Y direction after the collision between the corn seed and the test plate; $V_x$, $V_y$ are the velocities in the X direction and Y direction after the collision between the corn seed and the test plate; $g$ is the acceleration of gravity; $t$ is the time of the corn seed's oblique throwing motion.

During the test, the height of the backing plate $H_1$ and $H_2$ were changed, and the displacement $S_1$ and $S_2$ of the corn seeds corresponding to the two heights were measured, respectively, in the X direction, and the following formula could be obtained:

$$\begin{cases} V_x = \sqrt{\frac{g S_1 S_2 (S_1 - S_2)}{2(H_1 S_2 - H_2 S_1)}} \\ V_y = \frac{H_1 V_x}{S_1} - \frac{g S_1}{2 V_x} \end{cases} \tag{10}$$

where $S_1$, $S_2$, $H_1$, and $H_2$ are the X and Y displacements corresponding to the two tests after a collision between corn seed and test plate, respectively.

According to Formulas (9) and (10), the calculation formula of the collision recovery coefficient of corn seed can be obtained as follows:

$$\begin{cases} C_r = \frac{V_n{}'}{V_n} = \frac{\sqrt{(V_x{}^2 + V_y{}^2)}\cos\left[45° + \tan^{-1}\left(\frac{V_y}{V_x}\right)\right]}{V_0 \sin 45°} \\ V_0 = \sqrt{2gh} \end{cases} \tag{11}$$

where $C_r$ is collision recovery coefficient; $V_n{}'$ and $V_n$ are the normal rebound velocity after collision and the normal approach velocity before collision.

According to the above principle, corn seeds' collision recovery coefficient was determined using a contact parameter measuring device (The test device is shown in Figure 15). The material of the test plate and the backing plate was PMMA. The angle between the PMMA plate and the horizontal plane is fixed at 45°, the backing plate is fixed at the lower part of the frame, and the corn seed falling point height is 5 cm from the test plate. Replace the test plate with a prepared particle plate when measuring the corn–corn collision recovery coefficient.

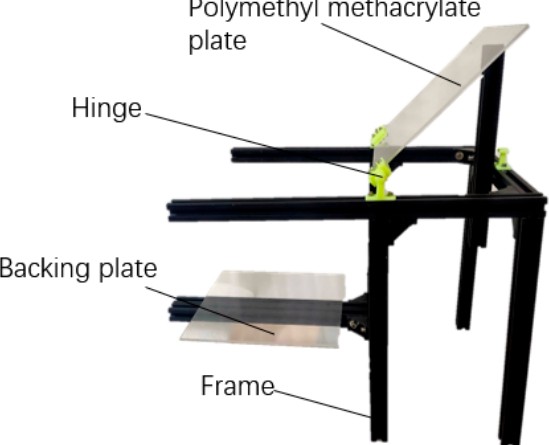

**Figure 15.** Collision recovery coefficient measuring device.

According to the above process, the two tests were repeated fifteen times, respectively, and the average value was obtained: the corn–PMMA collision recovery coefficient was 0.62, and the corn–corn collision recovery coefficient was 0.28.

### 3.4. Validation Test

3.4.1. Parameter Selection of Simulation Test

According to the classification of corn seeds in Section 3.1.3 and the measurement results of characteristic dimensions of these three types of corn seeds, the 3D model was established. The method of particle aggregate was used for modeling in EDEM2018 (Figure 16), and the Hertz–Mindlin with Bonding model was used for the contact model [27–29].

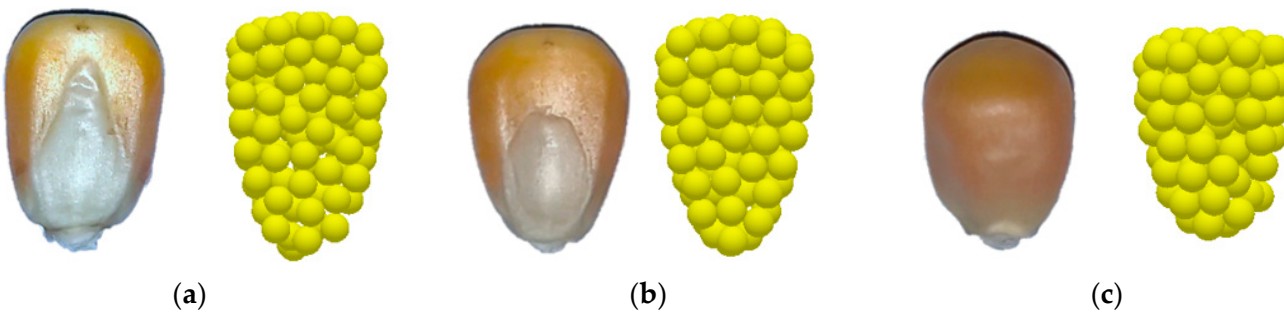

| (a) | (b) | (c) |

**Figure 16.** Simulation model of corn seeds: (**a**) flat corn seeds; (**b**) quasi-conical corn seeds; (**c**) quasi-cylindrical corn seeds.

Based on the above test results, the parameters required by the simulation test are shown in Table 5. All parameters in the table, except those obtained through experimental investigation in this study, were sourced from literature references [22].

**Table 5.** Parameters in DEM simulation.

| Parameters | Value |
| --- | --- |
| Density of corn seed/(kg·m$^{-3}$) | 1197 |
| Poisson's ratio of corn seed | 0.4 |
| Shear modulus of corn seed/Pa | $1.36 \times 10^8$ |
| Normal Stiffness of corn seed/(N·m$^{-3}$) | $3.54 \times 10^9$ |
| Shear stiffness of corn seed/(N·m$^{-3}$) | $2.53 \times 10^9$ |
| Critical normal stress of corn seed/Pa | $1.1 \times 10^7$ |
| Critical shear stress of corn seed/Pa | $4.1 \times 10^6$ |
| Bonded disk radius/mm | 1 |
| Density of PMMA/(kg·m$^{-3}$) | 1200 |
| Poisson's ratio of PMMA | 0.35 |
| Shear modulus of PMMA/Pa | $1.30 \times 10^9$ |
| Corn–corn rolling friction coefficient | 0.0784 |
| Corn–PMMA rolling friction coefficient | 0.0934 |
| Corn–corn collision recovery coefficient | 0.28 |
| Corn–PMMA collision recovery coefficient | 0.62 |
| Corn–corn static friction coefficient | 0.32 |
| Corn–PMMA static friction coefficient | 0.445 |

3.4.2. Plane Collision Test

In order to reduce the simulation test time, the plane collision method was used to carry out the verification test. In the simulation test, the contact parameters between the corn seed and the boundary are set to the above-measured contact parameters between the corn seeds. The results of the physical and simulation tests are shown in Figure 17. The experiment was divided into three groups, and each selected a corn seed type for

the plane impact test. Release the corn seed 15 cm from the particle plate and record the maximum height at which the corn seed bounces back after the first collision (Using the method in Section 3.3.1 to eliminate the effect of multiple impacts on the test results). The results of the three groups of experiments were averaged, and the maximum height of the three types of corn seeds rebound was 5.95 cm (flat), 7.68 cm (quasi-conical), and 7.79 cm (quasi-cylindrical), respectively. The above process was repeated in EDEM2018, and the maximum rebound heights of three types of corn seeds were 6.17 cm (flat), 7.41 cm (quasi-conical), and 7.53 cm (quasi-cylindrical), respectively. The relative errors of the simulation and physical tests are 3.70%, 3.52%, and 3.34%, respectively. The relative errors of the three groups of tests are less than 5%, which proves that the simulation test is reliable.

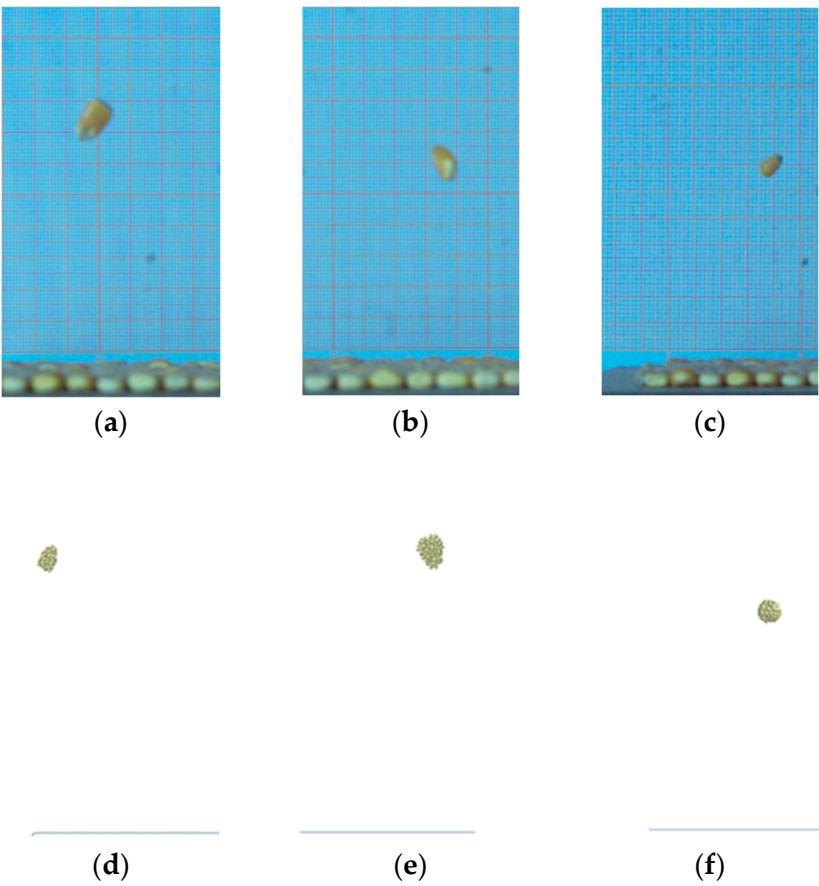

**Figure 17.** Plane collision test: (**a**) physical test of flat corn seed; (**b**) physical test of quasi-conical corn seed; (**c**) physical test of quasi-cylindrical corn seed; (**d**) simulation test of flat corn seed; (**e**) simulation test of quasi-conical corn seed; (**f**) simulation test of quasi-cylindrical corn seed.

### 3.4.3. Repose Angle Test

The repose angle test was carried out according to the method in Section 2.2. The particle pile formed by corn seeds was photographed (Figure 18a), then the photos were processed using Matlab, and the angle of the particle pile was obtained by linear fitting [30,31]. The above process was repeated fifteen times and obtained the average value. The repose angle of corn seeds was 31.38°.

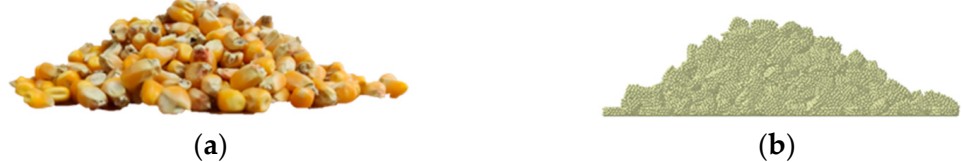

**Figure 18.** Repose angle test: (**a**) physical test; (**b**) simulation test.

After that, the three-dimensional model of the test device was imported into EDEM2018 software, and the parameters in Table 5 were substituted into EDEM for the simulation test. During the test, the number of corn seeds generated for each type was generated according to the final ratio in Section 3.1.3, and a total of six hundred corn seeds were generated. The simulation test was repeated fifteen times, and the average value was taken to obtain that the repose angle of corn seeds was 30.67°, the relative error between the simulated repose angle and the actual repose angle was 2.26%, and the relative error was less than 5%, indicating that the simulation test was reliable.

### 3.5. Discussion

(1)    The multi-point collision of corn seeds will reduce the rebound height of corn seeds, but the rebound height of corn seeds is affected by various factors, such as the collision angle between corn seeds and the falling posture of corn seeds. Therefore, the rebound height of corn seeds cannot be used as the basis for judging the multi-point collision between corn seeds.

(2)    After the collision between corn seeds, the more turns the corn seeds spin in the air, the lower the height of the corn seeds rebound.

(3)    When using the lifting method to conduct the repose angle test of corn seeds, the high-speed camera is used to shoot the test process. At the beginning of the experiment, when the cylinder was lifted, the corn seeds were dispersed in all directions after losing the barrier of the cylinder. Because no other object was around to block the corn seeds, they directly collided with the bottom surface to a small degree (Figure 19a). As the experiment continued, the corn seeds piled up on the bottom surface, and the seeds inside the cylinder came into contact with the fallen seeds. When the cylinder is lifted, the corn seeds in the original cylinder will mainly slide and roll due to the obstruction of the seeds below (Figure 19b).

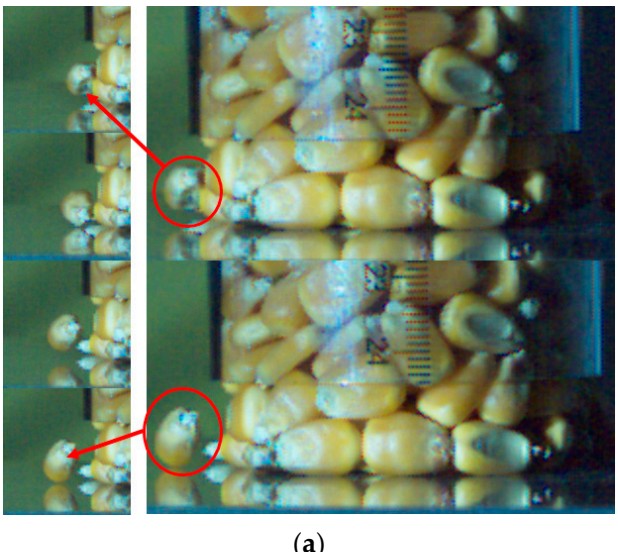

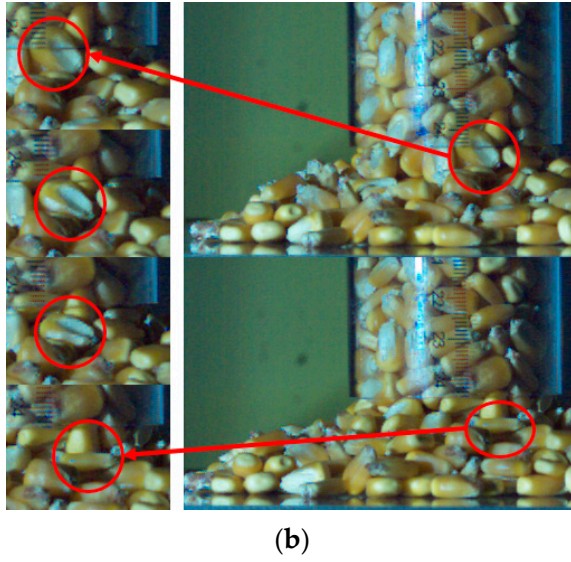

**Figure 19.** Movement of corn seeds in repose angle test: (**a**) Collision: The movement of corn seed in the red circle is shown on the left side of the picture; (**b**) Slide: The movement of corn seed in the red circle is shown on the left side of the picture.

## 4. Conclusions

In this study, corn seeds were first divided into three categories according to their surfaces, and the proportion of each type of corn seed was counted. Then, these three types of seeds were measured according to the defined characteristic dimensions. The relationship between the characteristic dimensions of corn seeds modified the statistical and measurement results. Then, corn seeds' static and rolling friction coefficients were measured using the inclined plane and energy conservation methods, respectively. When measuring the collision recovery coefficient of corn seeds, the multi-point collision of corn seeds is identified by combining high-speed photography and sound waveform to improve the accuracy of measurement results. The results are as follows:

(1) According to the shape of the surfaces of corn seeds, they can be divided into three categories: flat, quasi-conical, and quasi-cylindrical. The number of flat, quasi-conical, and quasi-cylindrical corn seeds accounted for 77.9%, 15.4%, and 6.7%, respectively.

(2) The sound waveform's peak value after the corn seeds' single-point collision is positive and remains stable in a specific time interval. The peak value of the sound waveform after the multi-point collision of corn seeds is zero and maintains a stable value within a specific time interval.

(3) Through physical tests, the corn–corn rolling friction coefficient and corn–PMMA rolling friction coefficient were 0.0784 and 0.0934, respectively. The corn–corn static friction coefficient and corn–PMMA static friction coefficient were 0.32 and 0.445, respectively. The corn–corn collision recovery and corn–PMMA collision recovery coefficients were 0.28 and 0.62, respectively.

(4) The measurements are verified by plane collision and repose angle tests. The relative errors between the simulation test and physical test of the two verification methods are less than 5%, which proves that the technique combining high-speed photography and sound waveform is reliable.

**Author Contributions:** Conceptualization, X.L. and W.Z.; methodology, X.L. and W.Z.; investigation, S.X., F.M., Z.D. and J.L.; resources, Y.M.; data curation, J.L.; writing, original draft preparation, W.Z. and X.L. All authors have read and agreed to the published version of the manuscript.

**Funding:** This research was funded by the National Natural Science Foundation of China (52275245) and Henan Science and Technology Research Program (222103810041).

**Institutional Review Board Statement:** Not applicable.

**Data Availability Statement:** The data used to support the findings of this study are available from the corresponding author upon request.

**Acknowledgments:** The authors would like to thank their college and the laboratory, as well as gratefully appreciate the reviewers and editors who provided helpful suggestions for this manuscript.

**Conflicts of Interest:** The authors declare no conflict of interest.

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
