# Peer review of "Calibration of Collision Recovery Coefficient of Corn Seeds Based on High-Speed Photography and Sound Waveform Analysis"

_agriculture, doi:10.3390/agriculture13091677_

Round 1

Reviewer 1 Report

Modify by referring to annotations. 

Revise and improve.

Author Response

The problems in the manuscript have been revised in light of the above valuable comments.

[Line 30-32] The statistical results of the corn planting area were added, and the content was improved.

[Line 232-235] The picture has been replaced with a clearer version. The reasons for using quasi-cylindrical corn seeds for experiments are also explained.

[Line 266-267] Moderate changes have been made to this section to make it easier to understand.

[Line 299-305] The processing of sound waveforms has been described in detail to make this part easier to understand.

[Line 381-384] This part has been modified, and the details of the simulation test are described.

Reviewer 2 Report

1. In the abstract, the numerical values ​​of the shape of corn seeds, sound waveform, friction coefficient and collision recovery coefficient should be indicated.

2. The word "static friction coefficient" should be added to the key words.

3. In Introduction: Paragraph 1 provides information on corn and Paragraph 2 provides information on EDEM programs. There is no connection between paragraph 1 and paragraph 2. It is necessary to ensure that they are interconnected.

4. In 2. Materials and Methods, we can see the result information of “The average moisture content of the three groups of corn seeds was 11.7%” in 2.1.1. Density and Moisture Content of Corn Seeds part and the result information of “The shear modulus of corn seed is 1.36×108 Pa” in 2.1.2. Determination of Poisson’s Ratio and Shear Modulus part. They should be moved to 3. Results and Discussion.

5. You can see similar cases in 2.1.3. Determination of Characteristic Dimensions of Corn Seeds, 2.2. Determination of Static and Rolling Friction Coefficient of Corn Seeds, 2.3. Determination of Collision Recovery Coefficient of Corn Seeds.

6. 3. Results and Discussion 3.2.1. Plane Collision Test and 3.2.2. Repose Angle Test sections contain information on experimental methods. They should be moved to 2. Materials and Methods.

Author Response

The problems in the manuscript have been revised in light of the above valuable comments.

  1. In the abstract, the numerical values ​​of the shape of corn seeds, sound waveform, friction coefficient and collision recovery coefficient should be indicated.

Response: The above content has been added by an appropriate modification to the abstract.

  1. The word "static friction coefficient" should be added to the keywords.

Response: The word "static friction coefficient" has been added to the keywords.

  1. In Introduction: Paragraph 1 provides information on corn and Paragraph 2 provides information on EDEM programs. There is no connection between paragraph 1 and paragraph 2. It is necessary to ensure that they are interconnected.

Response: The content of the second paragraph has been modified to enhance the correlation between the first and second paragraphs.

  1. In 2. Materials and Methods, we can see the result information of “The average moisture content of the three groups of corn seeds was 11.7%” in 2.1.1. Density and Moisture Content of Corn Seeds part and the result information of “The shear modulus of corn seed is 1.36×108 Pa” in 2.1.2. Determination of Poisson’s Ratio and Shear Modulus part. They should be moved to 3. Results and Discussion.

Response: The order and structure of the contents of the above chapters have been revised.

  1. You can see similar cases in 2.1.3. Determination of Characteristic Dimensions of Corn Seeds, 2.2. Determination of Static and Rolling Friction Coefficient of Corn Seeds, 2.3. Determination of Collision Recovery Coefficient of Corn Seeds.

Response: The order and structure of the contents of the above chapters have been revised.

  1. 3. Results and Discussion 3.2.1. Plane Collision Test and 3.2.2. Repose Angle Test sections contain information on experimental methods. They should be moved to 2. Materials and Methods.

Response: The order and structure of the contents of the above chapters have been revised.

Based on the above recommendations, appropriate changes have been made to the content and structure of the sections "Materials and Methods" and "Results and Discussion."
